# REVISITING CRITICAL LEARNING PERIODS IN DEEP NEURAL NETWORKS

## ABSTRACT

Deep neural networks (DNNs) exhibit critical learning periods (CLPs) during early training phases, when exposure to defective data can permanently impair model performance. The prevalent understanding of such periods, primarily based on the interpretation of Fisher Information (FI), attributes CLPs to the memorization phase. However, our theoretical and empirical study exhibits that such explanations of CLPs are inaccurate because of the misunderstanding of the relationship between FI and model memorization. As such, we revisit the CLPs in DNNs from the information theory and optimization perspectives, gaining a better and more accurate understanding of CLPs. We visualize model memorization dynamics and observe that CLPs extend beyond the memorization phase. Additionally, we introduce the concept of the effective gradient, a novel metric able to quantify the actual influence of each training epoch on the optimization trajectory. Our empirical and theoretical analyses reveal that the norm of effective gradients generally diminishes over training epochs and eventually converges to zero, highlighting the disproportionate larger impact of initial training on final model outcomes. Besides, this insight also clarifies the mechanism behind permanent performance degradation due to defective initial training: the model becomes trapped in the suboptimal region of parameter space. Our work offers novel and in-depth understandings of CLPs and sheds light on enhancing model performance and robustness through such periods.

## 1 INTRODUCTION

The critical period refers to a specific time window in the early post-natal development of humans and animals during which individuals are exceptionally sensitive to certain external stimuli or experiences (Kandel et al., 2000; Wiesel & Hubel, 1963; Wiesel, 1993; Konishi, 1985). Such a period is crucial for the lifelong development of specific abilities or behavioral patterns. Inappropriate stimulation or experiences during this period can lead to permanent impairment of a skill. Inspired by the parallels between deep neural networks (DNNs) and biological neural connections, Achille et al. (2018) first observed a similar phenomenon in the training process of neural networks. Specifically, if the model is trained with defective data, such as blurred data, during this initial period, called critical learning periods (CLPs), it will suffer permanent impairment in its final performance, regardless of any additional training with high-quality data successively. Thereafter, similar CLPs phenomena have been observed in deep linear networks (Kleinman et al., 2023a), under multi-view setting Kleinman et al. (2023b), and under the federated learning paradigm (Yan et al., 2022; 2023a;b). These periods are demonstrated to be crucial windows for potentially enhancing model performance and lifting robustness against attacks Yan et al. (2022; 2023a;b).

To date, the underlying mechanisms that induce the CLPs and their resulting permanent impairment in DNNs have not been fully explored. The predominant explanations are based on observations related to Fisher Information (FI), as reported in Achille et al. (2018). It was noted that the FI metric increases during the initial training periods and subsequently decreases to a certain level and a significant increase in FI is observed when the model undergoes defective training at an early stage. They interpret the dynamics in FI as indicative of a "memorization phase," when the model memorizes information from the training data, followed by a "forgetting phase," during which the model reduces its retained information. Then, the observation of FI surge during defective training is attributed to abnormal growth in "synaptic strength" of neural connections (Achille et al., 2018) as the model memorizes too much defective data, preventing the neural connections from adapting

themselves effectively to the normal data afterward under high strength. Therefore, they conjecture that CLPs are centered within this memorization phase.

However, the prevailing narrative about FI-based model memorization and its correlation with CLPs presents significant issues. First, it mistakenly conflates CLPs—an intrinsic characteristic of the model during training—with observations of model impairment caused by defective data and the corresponding FI dynamics. The performance impairment from defective training is the result of interference with CLPs, with the surge in FI as a consequence of unstable learning induced by defective data, not the cause of CLPs. Therefore, using FI dynamics during defective training to explain CLPs confuses cause and effect. Second, FI on training data does not accurately quantify how much information a model retains but instead measures the model's sensitivity to the training data and how much the training data contributes to the model training process. A rise in FI likely reflects increased sensitivity to noisy information in the defective data, rather than indicating the memorization phase. As such, the claim that CLPs are driven by memorization based on FI dynamics warrants further scrutiny. Given these considerations, a re-examination of CLPs in light of these dynamics is essential.

In this paper, we conduct an in-depth exploration of the CLPs in DNNs from the perspectives of information theory and optimization. First, we provide a theoretical analysis of FI, clarifying its correlation with defective training and its inadequacy in analyzing model memorization. Following Shwartz-Ziv & Tishby (2017), we employ mutual information, which directly measures the information shared between two variables, to visualize the model's memorization phase and demonstrate that CLPs are not solely centered in this phase. Second, we shift our focus to the fundamental aspects of model training, i.e., SGD (Stochastic Gradient Descent) optimization essence. We propose a metric, called the *effective gradient*, to measure the contribution of the update in each epoch toward the optimization objectives. We provide theoretical proof that the norm of the effective gradient will converge to zero, exhibiting a decreasing trend, and conduct extensive experiments to validate its variations in practice. Consequently, the initial training period naturally exerts a larger contribution to the parameter updates during the optimization process, dictating the final performance level and illustrating the phenomenon of initial CLPs in the model. Additionally, the irrecoverable deterioration of effective gradients following initial defective training makes the model unlikely to escape from the sub-optimal area, leading to permanent performance impairment.

The contribution of this paper can be summarized as follows:

- We conduct a rigorous theoretical analysis and empirical reassessment of FI, unveiling significant misunderstandings for its indicative to CLPs. Our analysis is crucial as it corrects the foundational motivations that have driven past research on CLPs.

- We explain the CLPs from the perspective of the optimization process, providing a deeper understanding of why the initial training stages are pivotal. We propose the effective gradients metrics and mathematically reveal that the effective gradients decline in the model optimization dynamics, incurring the observation of CLPs.

- We envision several key perspectives that are vital for guiding future research on improving model robustness and security by leveraging a more informed understanding of CLPs. These new perspectives expand the scope of CLPs research, paving the way for innovative approaches to enhance model training performance.

## 2 EXISTING EXPLANATIONS TO CLPs

### 2.1 BACKGROUND OF FISHER INFORMATION (FI)

Consider a neural network $f(\cdot)$ parameterized by the parameter $\theta$, and denote the output probability of class $y$ given input $x$ as $p_\theta(y|x)$. If the weights are perturbed by $\delta\theta$, the new weights are $\theta' = \theta + \delta\theta$, and the perturbed output distribution is $p_{\theta'}(y|x)$. Here, we focus on the discrepancy between $p_\theta(y|x)$ and $p_{\theta'}(y|x)$, by calculating their KL (Kullback-Leibler) divergence and approximating the perturbed output distribution $p_{\theta'}(y|x)$ via Taylor expansion of $p_\theta(y|x)$ around $\theta$, yielding

$$KL(p_\theta \parallel p_{\theta'}) = \mathbb{E}_x \left[ \int p_\theta(y|x) \log \frac{p_\theta(y|x)}{p_{\theta'}(y|x)} \, dy \right] \approx \frac{1}{2}\delta\theta^T \cdot F(\theta) \cdot \delta\theta + o(\delta\theta^2) \,,$$

where $o(\delta\theta^2)$ includes higher-order small terms and $F$ is the FI Matrix (*FIM*), calculated by

$$F(\theta) = \mathbb{E}_{x \sim \hat{Q}(x)} \left[ \mathbb{E}_{y \sim p_\theta(y|x)} \left[ \nabla_\theta \log p_\theta(y|x) \nabla_\theta \log p_\theta(y|x)^T \right] \right] . \tag{1}$$

Essentially, FIM captures how sensitive the model output probabilities are to changes in the parameter across the data distribution (Amari & Nagaoka, 2000). In practice, since FIM is too large to compute, we usually adopt the trace of FIM, denoted as $\text{Tr}(F(\theta))$, to represent its value (Achille et al., 2018), which is abbreviated as FI in the rest of this paper.

## 2.2 INTERPRETATION OF CLPS THROUGH FI

To explore the CLPs, Achille et al. (2018) proposed to observe the FI dynamics in the training. First, during the training process, the Fisher Information (FI) of the training data is observed to initially increase in the early stages and then decrease to a stable level. The authors state that FI can be interpreted as a measure of the amount of information about the training data that the model retains. Based on this, the fluctuation in FI is taken to suggest that the model first enters a "memorization phase", where it memorizes information from the training data, followed by a "forgetting phase", where redundant or irrelevant information is discarded. Consequently, the authors conjecture that Critical Learning Periods (CLPs) are concentrated in the initial memorization phase, proposing that an FI increase could be used as an indicator for detecting these periods.

Second, if the model is trained on defective data during the early epochs, an abnormally high FI on the training data value results. The authors argue that when the training data is severely corrupted in the early stages, the network is forced to memorize more information to make predictions, which substantially increases the model's neural connection strength, as reflected by the high FI values. As a result, even if normal data is provided later, the network burdened by overly strong connections may struggle to adjust its connectivity, as described in Kirkpatrick et al. (2017), leading to impaired performance in the final model.

## 2.3 FLAWS IN FI-BASED EXPLANATION

The aforementioned explanations suffer two significant flaws, illustrated below. First of all, the authors mistakenly conflate Critical Learning Periods (CLPs)—an intrinsic characteristic of the model during training—with observations of model impairment caused by defective data and the corresponding Fisher Information (FI) dynamics. CLPs should be understood as the initial phase during which the model's learning disproportionately impacts its final performance, regardless of the training data quality, thereby should be explained through the dynamics of normal training. The performance impairment observed after training on defective data is a result of interference with CLPs, with changes in FI merely reflecting this disruption, not explaining CLPs itself. Therefore, using FI dynamics to directly explain the existence or behavior of CLPs is a case of confusing cause and effect: the surge in FI is a consequence of the model's unstable learning from defective data rather than a fundamental driver of CLPs. While FI on the training data and CLPs may be correlated, they are not causally linked.

Moreover, FI on the training data cannot accurately quantify the amount of information a model retains about the training data. Instead, it measures the model's sensitivity to the data and the amount of information that the training data can contribute to the model training. A rise in FI likely indicates that the model is becoming increasingly sensitive to noisy information abundant in the defective training data, rather than entering a memorization phase. As a result, the claim that CLPs are centered on the memorization phase based on FI dynamics observation warrants further examination.

## 3 REVISITING FI DYNAMICS AND MODEL MEMORIZATION

This section first theoretically explores the dynamics of FI. Then we examine the relationship between the CLPs and the model memorization to answer whether the CLPs are centered in the model memorization phase.

### 3.1 THEORETICAL ANALYSIS OF FI

Achille et al. (2018) attributes the increase in FI to the rise of data information contained in the model and proposes to observe the initial CLPs through FI. However, it's important to correct that the

definition of FI is instead the amount of information that an observable random variable $X$ carries about an unknown parameter $\theta$ of a distribution that models $X$ (Fisher, 1925). In the machine learning context, $X$ is the training data, and $\theta$, being the distribution that models $X$, is the model parameter, so FI measures the amount of information provided by training data about the model parameters. A mathematical interpretation is provided below.

**Proposition 1.** *The larger the $F(\theta)$ on data $X$, the lower uncertainty of the model parameter $\theta$ that is estimated from data $X$.*

*Proof.* The proof is based on the well-known Cramér-Rao Lower Bound (Cramér, 1999; RAO, 1945). Define the variance of any unbiased estimator $\hat{\theta}$ as $Var(\hat{\theta})$, bounded by the reciprocal of the FI, i.e.,

$$Var(\hat{\theta}) \geq 1/F(\theta) . \tag{2}$$

Hence, a larger $F(\theta)$ implies a tighter lower bound on the variance of model parameter $\theta$. This directly means less uncertainty in estimating model parameter $\theta$ from data $X$. $\square$

According to proposition 1, we can comprehend why FI is aptly termed "information," as it quantifies the uncertainty degree in estimating model parameters via data $X$. FI of training data essentially quantifies the amount of information the training data provides for optimizing the model parameters, rather than the information the model contains about the training data. Moreover, FI of the training data decreases during normal training as follows:

**Theorem 1.** *If the model is trained with SGD with necessary assumptions, as presented in Appendix A.1, holds and the learning rate $\eta < \frac{2}{M}$, we have:*

$$\lim_{t \to \infty} Tr(F(\theta_t)) = 0 , \tag{3}$$

*where $\theta_t$ is the model parameter under epoch $t$ and $M$ is the M-Lipschitz parameter as shown in Appendix A.1.*

The proof is deferred to Appendix A.2. Theorem 1 demonstrates that the Fisher Information (FI) of the training data in a model trained using SGD will eventually converge to zero, indicating that the model parameters stabilize as training progresses. This aligns with Proposition 1, as the model continues to train on the data and becomes more well-generalized, the remaining information the training data provides for optimizing the model parameters decreases. If we equate model memorization with the dynamics of FI during the normal training process, an unreasonable conclusion will be led: the model does not memorize anything. This contradicts the reality that a model can still memorize training data, even when its parameters stabilize and FI decreases. Therefore, while FI offers insight into the information the training data provides for model optimization, it is insufficient to describe or quantify the model's memorization behavior.

### 3.2 MODEL MEMORIZATION MEASUREMENT

The two-phase learning phenomenon in neural networks, where a model initially memorizes training data and then forgets label-irrelevant information, has been observed in simple neural networks (e.g., MLPs) through the *Mutual Information* dynamics (Shwartz-Ziv & Tishby, 2017). Here, we follow their work to adopt the mutual information metrics to reexamine the relationship between the memorization phase and the CLPs.

Given any two random variables $X$ and $Y$, with a joint distribution $p(x, y)$, their Mutual Information is defined as:

$$I(X; Y) = D_{KL}[p(x, y)\|p(x)p(y)] = H(X) - H(X|Y) ,$$

where $D_{KL}[\cdot]$ measures the KL divergence and $H(\cdot)$ is the entropy. It measures the reduction in uncertainty about $X$ due to the knowledge of $Y$, thereby quantifying the information that $Y$ contains about $X$. In the machine learning context, we can consider the neural network parameterized with $\theta$, denoted by $f(;\theta)$, as an encoder. For the training dataset $X$, the model outputs of $T_t = f(X; \theta_t)$ represent the learned representations from training data. The mutual information of $I(X; T_t)$ measures the quantity of information that learned representations contain about the training data, specifically, the information memorized by the model in epoch $t$. Thus, by analyzing the evolution of $I(X; T_t)$ as the training epoch grows, we can understand how the model memorization phase happens, further exploring its relationship to CLPs. Although advanced methods for approximating

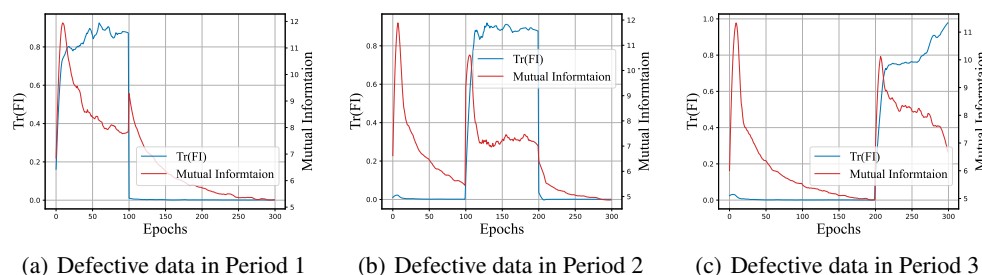

| (a) Defective data in Period 1 | (b) Defective data in Period 2 | (c) Defective data in Period 3 |

Figure 1: The mutual Information and the FI variations on ResNet-18.

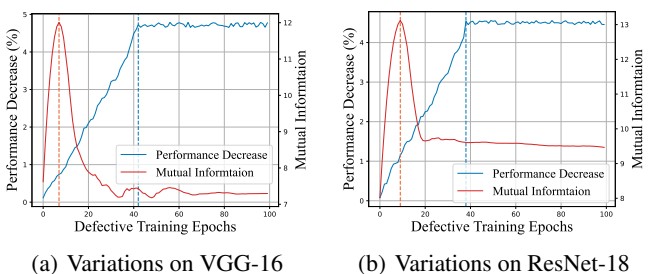

| (a) Variations on VGG-16 | (b) Variations on ResNet-18 |

Figure 2: Variations in performance decrease and mutual information during defective training.

precise mutual information values exist, as proposed in Kleinman et al. (2021); Belghazi et al. (2018), our primary objective is to observe the dynamics of mutual information. Therefore, we calculate mutual information using the binning method, which is widely adopted in Shwartz-Ziv & Tishby (2017); Saxe et al. (2019); Goldfeld et al. (2019).

### 3.3 EXPERIMENT ANALYSIS

Given the above understanding, we experimentally revisit the FI explanation from Achille et al. (2018). We aim to explore three questions: 1) whether FI phenomenon is the consequence of defective training; 2) whether an FI variation indicates the model memorization phase; and 3) whether the CLPs are centered in the memorization phase.

Since Achille et al. (2018) has demonstrated that the CLPs and FI dynamics do not rely on any specific learning rate, batch size, model structure, and datasets, we conduct experiments with general settings. Here, we exhibit the experiments for ResNet-18 training on CIFAR-10, as the approximation of the FI is more stable for ResNet-18 with a smooth loss landscape (Li et al., 2018). We employ the general SGD as the optimizer with a fixed learning rate of 0.01 and a total training epochs of 300. Following the methodology described in Achille et al. (2018), defective data are created by applying heavy Gaussian blur to the original training dataset, and FI is quantified as the trace of the FI Matrix under Monte-Carlo sampling approximation. Initially, we train the models using standard training data, achieving baseline testing accuracies of 91.34% for VGG-16 and 93.28% for ResNet-18, respectively. For the defective training, we divide the training epochs into three equal periods: Period 1 (epochs 1-100), Period 2 (epochs 101-200), and Period 3 (epochs 201-300). We train three models where defective data is used as the training data for one period in each model, while the remaining two periods utilize the original dataset. Detailed settings, metric calculations, and the results for VGG-16 are provided in Appendices B.1, B.2, and B.3.

**Finding 1: The surge in FI is the consequence of defective training:** Figure 1 illustrates the variations in mutual information and FI on ResNet-18. We can observe that an FI surge can occur at any point during training once the model is exposed to defective data, as evidenced by the increases in the blue lines during any defective training periods in Figure 1. This surge occurs because the model becomes more sensitive to noise and spurious patterns present in the defective data, leading to an increase in FI. In other words, despite being unreliable, these defective data still provide a significant amount of noisy information to the model. This suggests that the fluctuations in FI are merely a consequence of defective training, rather than a fundamental explanation for the Critical Learning Period (CLPs) that occurs during the early stages of normal training. While FI captures the

instability caused by noisy or low-quality data, it does not directly explain the model's ability to learn meaningful patterns during the CLPs.

**Finding 2: FI dynamics do not match the model memorization variation:** First, from Figure 1, we can observe that during the training periods with normal data, FI rapidly decreases to near zero. For example, during epochs 0-200 in the settings shown in Figure 1(c), the model undergoes a normal training process with the original dataset, achieving a test accuracy of 93.29%, while FI eventually approaches 0.0026. This indicates that such a FI trending does not reflect that the model has been well-trained and has memorized useful knowledge from the training data. Additionally, we observe that mutual information increases rapidly when the model encounters new data, regardless of whether it is defective or normal, and eventually decreases to a certain value larger than 0. This suggests that the model indeed goes through a memorization phase and then forgets label-irrelevant information to retain the useful knowledge.[1] It is consistent with the findings in Shwartz-Ziv & Tishby (2017). However, while mutual information begins to decrease during the defective training epochs, FI continues to increase, as evidenced by the rise in the blue line while the red line drops during each defective training period. This discrepancy occurs because the model starts to forget label-irrelevant information while the defective data is too poor to learn from, causing the model to remain unstable, which is reflected by the increasing FI. These observations demonstrate that FI does not accurately capture the model's memorization phase.

**Finding 3: CLPs do not center on the memorization phase:** Figure 1 also illustrates that the model memorization phase can occur at any training period when the model is exposed to new data, as evident from the multiple increases in the mutual information curves. Interestingly, after training with defective data in Period 2 and recovering with original data in Period 3, the final accuracy of VGG-16 and ResNet-18 models is 91.46% and 93.26%, respectively, comparable to the baseline performance of 91.34% and 93.28%. This suggests that memorizing defective data during the middle memorization phase is reversible and does not lead to permanent damage to final model performance. In contrast, training with defective data during the CLPs leads to permanent impairment, highlighting the distinction between the CLPs and the memorization phases. Additionally, we also conduct the experiment to reveal the variations in model performance impairment and mutual information under defective training epochs, during which the model is trained on defective data, as depicted in Fig. 2. The dotted lines in the figure indicate the epochs at which maximum mutual information and model performance impairment occur. We observe that for VGG-16 and ResNet-18, the impairment of final model performance reaches its maximum at epochs 42 and 38, respectively, suggesting that the CLPs should conclude around these epochs. However, mutual information peaks earlier, at epochs 8 and 7, indicating that the memorization phase ends much earlier. Therefore, the CLPs are even unaligned with the initial model memorization phase. In conclusion, the CLPs are not simply centered in the memorization phases as previously thought, calling for a new explanation.

## 4 EXPLAINING CLPs THROUGH OPTIMIZATION

As discussed in Section 3, the current narrative that explains CLPs through FI dynamics during the defective training period encounters significant limitations and issues. Hence, we pivot to the essence of neural network training, i.e., the optimization process, and offer a novel perspective for theoretically analyzing why the initial training epochs can dominate the overall performance of a model undergoing general training, i.e., behaving as CLPs.

### 4.1 STOCHASTIC GRADIENT DESCENT

The mainstream model training is essentially an optimization process in which model parameters are updated according to gradient descent rules. Among different popular optimization algorithms, such as momentum and Adam (Kingma & Ba, 2014; Loshchilov & Hutter, 2018), their foundation lies in Stochastic Gradient Descent (SGD) (Robbins & Monro, 1951). Consider a neural network parameterized by $\theta_t$ in epoch $t$, represented as $f(;\theta_t)$, its update process under SGD is governed by the following iterative formula:

$$\theta_{t+1} = \theta_t - \eta\nabla_{\theta_t}\ell(\mathbf{x};\theta_t;\xi) , \tag{4}$$

---

[1]However, the reason why and when the model spontaneously forgets label-irrelevant information remains unclear (Shwartz-Ziv & Tishby, 2017) and is beyond the scope of this paper.

where $\nabla_{\theta_t}\ell(\mathbf{x}; \theta_t; \xi)$ is the stochastic gradient calculated with the subset of the whole training dataset, which incurs a noise $\xi$. Following this equation, the model updates its parameter to align with the negative gradient direction in each iteration. Besides, to ensure the convergence of the SGD, in particular, we hold some necessary assumptions (Nemirovski et al., 2009; Li & Orabona, 2019), which are presented in Appendix A.1

## 4.2 THEORETICAL ANALYSIS OF GRADIENT DYNAMICS

The CLPs refer to the phenomenon where the initial training phase has a disproportionately significant influence on the final model performance. In other words, the impact of model gains on its final performance decreases as training progresses, making any defects acquired during the initial period difficult to recover from and potentially leading to permanent impairment. To explore this phenomenon, our motivation is to identify a metric able to capture this diminishing impact. As suggested by Eqn. (4), if the learning rate $\eta$ is a constant, which is small enough to ensure convergence, the magnitude of model updates is modulated by the stochastic gradient, particularly its norm. Hence, we aim to demonstrate that the influence of the stochastic gradient dynamics on the model's way to the optimum decreases throughout the optimization process, confirming that the initial training epochs have a greater impact on the final performance.

Considering the random noise introduced in the stochastic gradient of each training epoch will inevitably result in the uncertainty of its norm, we link the expected norm of the stochastic gradient to that of the true gradient, denoted as $\ell(\theta)$. The true gradient is defined as the gradient computed based on the entire data samples, without the noise caused by random sampling.

**Lemma 1.** *The expectation norm of stochastic gradient is bounded as*

$$\mathbb{E}[||\nabla\ell(\theta; \xi)||^2] \le \mathbb{E}[||\nabla\ell(\theta)||^2] + \sigma^2 . \tag{5}$$

The proof is deferred to Appendix A.3. Lemma 1 indicates that the expectation norm of the stochastic gradient is bounded by the expectation norm of the true gradient and the variance of the stochastic gradient noise. As such, we can further explore its expectation norm's variation via the true gradient.

**Lemma 2.** *If learning rate $\eta < \frac{2}{M}$, the expected norm of the stochastic gradient $\mathbb{E}[||\nabla\ell(\theta; \xi)||^2]$ will converge alongside the true gradient $\mathbb{E}[||\nabla\ell(\theta)||^2]$, that is*

$$\mathbb{E}[||\nabla\ell(\theta)||^2] \to 0, \ \mathbb{E}[||\nabla\ell(\theta; \xi)||^2 \to \sigma^2 \tag{6}$$

The proof is deferred to Appendix A.4. According to Lemma 2, the expected norm of the stochastic gradient converges alongside the expected norm of the true gradient, ultimately reaching the variance of the noise as the true gradient approaches zero. In other words, it implies that the expected norm of the stochastic gradient will not reach zero; even at the end of the training process, stochastic gradients persist. As a result, the stochastic gradient can not precisely reflect the trajectory to the optimum during the optimization process due to its noisy nature. Given this understanding, to more accurately measure the actual magnitude of model updates in each epoch, we propose a new metric called the effective gradient, denoted as $\hat{\ell}(\theta; \xi)$ and defined as below:

**Definition 1.** *The effective gradient is defined as the projection of the stochastic gradient on the direction of the true gradient, that is*

$$\nabla\hat{\ell}(\theta; \xi) = Proj_{\nabla\ell(\theta)}\nabla\ell(\theta; \xi) . \tag{7}$$

The norm of the effective gradient can reflect the actual contribution of the model update along the direction to the optimum brought by the stochastic gradient. Thus, it can reveal the true contribution of the stochastic gradient in achieving the optimization objectives in each epoch. Upon Lemmas 1 and 2, we have the following theorem for analyzing its variations.

**Theorem 2.** *If learning rate $\eta < \frac{2}{M}$, the expected norm of the effective gradient $\mathbb{E}[||\hat{\ell}(\theta; \xi)||^2]$ converges alongside the true gradient $\mathbb{E}[||\nabla\ell(\theta)||^2]$, that is*

$$\mathbb{E}[||\nabla\ell(\theta)||^2] \to 0, \ \mathbb{E}[||\nabla\hat{\ell}(\theta; \xi)||^2] \to 0 , \tag{8}$$

*where $M$ is the M-Lipschitz parameter as shown in Appendix A.1*

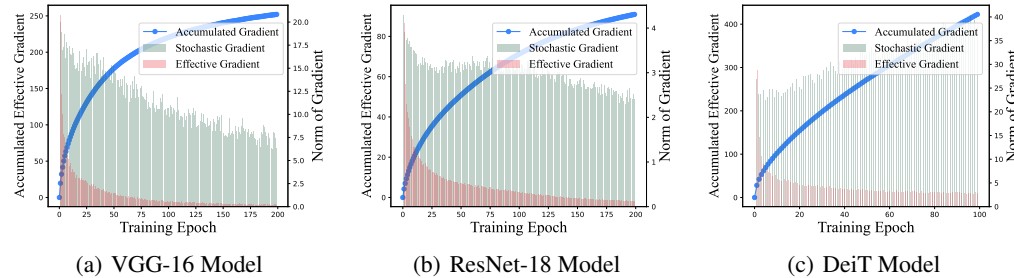

| (a) VGG-16 Model | (b) ResNet-18 Model | (c) DeiT Model |

Figure 3: The variation of the norm of effective gradient, stochastic gradient, and the accumulated effective gradient with normal training.

The proof is presented in Appendix A.5. Theorem 9 suggests that the expected norm of the effective gradient also converges alongside the expected norm of the true gradient, ultimately reaching zero. Although the non-convexity of the loss function, in general, prevents the expected norm of the true gradient from decreasing monotonically, Theorem 9 still implies that the norm of the effective gradient is in the decreasing tendency throughout the entire training process. We will provide the empirical results to exhibit this phenomenon in Section 4.4.

### 4.3 EXPLAINING CLPS THROUGH EFFECTIVE GRADIENT

Theorem 9 provides an important observation: as the training epoch progresses, the contribution of model updates, i.e., the norm of the effective gradient, to achieve the optimization objectives diminishes. This gives a hint that the initial training period naturally has a larger impact on the parameter update during the optimization process in dominating the final performance, thereby behaving the phenomenon that the model possesses initial CLPs. If the model is trained on defective data during initial epochs, the resulting updates will be misguided but significant, causing the model to converge to a suboptimal parameter space rapidly. Even if subsequent training on good data occurs, the actual optimization effect of the model updates gained will have a much lower influence compared to the early defective data, making it difficult for the model to escape from this suboptimal space. As a result, the model may suffer permanent performance impairment.

### 4.4 EXPERIMENT ANALYSIS

In this section, we evaluate the effective gradient variation during the training process to explore the following questions: 1) whether the expected norm of effective gradient behaves the decreasing trend; 2) what causes the model permanent impairment?

To justify the versatility of the effective gradient in probing the critical learning phase, our experiments on CIFAR-10 encompass both the conventional model architectures, such as VGG-16 and ResNet-18, and the transformer architecture DeiT. The optimizer adopted is SGD with a fixed learning rate with configurations detailed in Appendix B.1. Note that we fixed the learning rate rather than using an annealing scheme to demonstrate that the decrease of the effective gradient during the optimization process does not result from the annealed learning rate. The experimental results on models trained on other popular optimizers are presented in Appendix B.3. The training epochs are 200 for ResNet and VGG, and 100 for DeiT, enough for the model to converge. The defective data is generated by applying heavy Gaussian blur to the original training dataset, as documented in Achille et al. (2018).

The calculation of the effective gradient depends on the actual gradient, which is derived from the entire dataset at once and is hard to measure in practice. Instead, considering the essence that the effective gradient is to measure the actual contribution of the model update along the direction to the optimum, we use the vector pointing from the current model parameters to the optimum to approximate the true gradient. Here, to reduce the approximation error, we take the parameters averaged from the model with top-10 testing accuracy as the optimum. Then, we calculate the projection of the stochastic gradient in this approximated direction as the effective gradient. Besides, the effective gradients are always calculated on the current training data, thus during the defective/normal training period, the gradients are obtained upon defective/original data.

**Finding 4: The dynamics of effective gradient reflect the CLPs**: Figure 3 exhibits the variation in the norms of the effective gradient, the stochastic gradient, and the accumulated effective gradient.

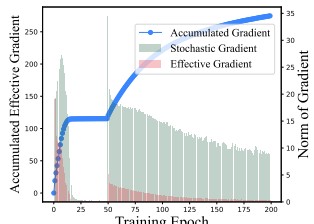 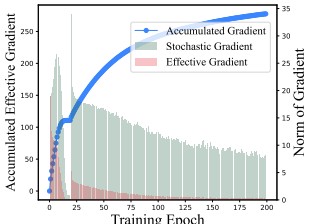 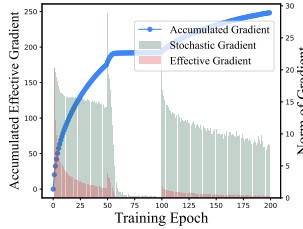

(a) 50 epochs defective training at initial stage, leading to 5.78% accuracy decrease

(b) 25 epochs defective training at initial stage, leading to 4.92% accuracy decrease

(c) 50 epochs normal training at initial stage followed by 50 epochs defective training, merely leading to 0.17% accuracy fluctuation

Figure 4: The variation of the norm of effective gradient, stochastic gradient, and the accumulated effective gradient with defective training.

We observe that the variation in these gradients aligns with our theoretical analysis: the stochastic gradient tends to converge to the variance of the noise, while the effective gradient approaches zero. Notably, the norm of the effective gradient exhibits a pronounced decreasing trend during the training epochs, characterized by an initial sharp decline in its red bar across all model structures. This pattern supports our assertion that the actual contribution of the model update in the direction of the optimum decreases, resulting in the updates during the initial epochs contributing significantly more to the optimization process during the whole training. Additionally, the blue line representing the accumulated effective gradient showcases a rapid increase initially, followed by a plateau, further indicating that in the initial epochs, the model has already made significant progress toward the optimum model. Indeed, it is this disproportionate large contribution to model optimization during the initial epochs that induces us to observe the CLPs in the early stages of training.

**Finding 5: Unrecoverable effective gradients deterioration after defective training leads to the impairment**: Next, we utilize the effective gradient to elucidate the permanent impairment resulting from defective training during the CLPs. First, Figure 4(a) displays the variation of different gradients of the model that is trained with defective data in the initial 50 epochs, the CIFAR-10 dataset is employed as the example. We can observe that the norm of effective gradient rapidly decreases and converges to 0 after 25 epochs, indicating that the model parameters reach a basin, albeit sub-optimally due to training on defective data. Moreover, after the 50-th epoch, although the model is exposed to normal data and experiences a significant stochastic gradient, as indicated by the tallest green bar, the actual progress towards the optimum remains limited, as shown by the short red bar representing the norm of the effective gradient. This suggests that gains from the stochastic gradient after the initial defective training are substantially reduced, which may enable the model to escape from the bad basin induced by defective data but are insufficient to propel the model toward the optimal performance, leading to 5.78% accuracy impairment. Even if we shorten the defective training period to 25 epochs at which the model just converges to the sub-optimal, Figure 4(b) still exhibits the similar phenomenon that unrecoverable deteriorated gains on the effective gradients, as shown by the short and decreasing red bar representing the norm of the effective gradient, and the model undergoes 4.92% accuracy impairment.

**Finding 6: Defective training after CLPs will not lead to permanent impairment:** Figure 4(c) displays the variation in different gradients of the model that is initially trained with normal data for 50 epochs, followed by 50 epochs of defective training. During the initial critical period, the model optimizes its parameters through the normal training process, which is accompanied by smooth variations in both stochastic and effective gradients. This process is interrupted by the onset of defective training, as indicated by the sharply increased stochastic gradients. However, the deterioration of the effective gradient instead acts as a safeguard, preventing the model from significantly drifting due to the defective data, as demonstrated by the short and rapidly decreasing red bar of the effective gradient and the slight increase in the blue line of the accumulated effective gradient. Therefore, after the defective training period, the model can readily correct the adverse movements and return to a favorable optimization trajectory, yielding merely 0.17% accuracy fluctuation.

In summary, the variation in effective gradients demonstrates a robust indication for theoretically explaining and empirically interpreting the CLPs at the initial stage of training.

## 5 Discussions

Based on our analysis, we challenged the predominant explanation of CLPs and offered an optimization-based perspective for theoretically explaining them. Our exploration provides valuable insights into understanding DNNs and leaves problems that should be explored in the future.

**Critical learning periods are not time windows with clear boundaries.** According to our analysis, the importance of the initial training period is attributed to a decreasing contribution gained in each epoch throughout the training process. This implies that the CLPs are the external manifestation of this consistently decreasing trend. In other words, an early training epoch is always more important than its subsequent ones, rather than some certain initial epochs within time windows signify the CLPs. Additionally, Figures 4(a) and 4(b) reveal why model impairment does not infinitely increase but instead reaches its maximum at certain epochs: the model becomes trapped in a sub-optimal valley determined by the defective data after a certain number of epochs. Therefore, existing studies Achille et al. (2018); Yan et al. (2022) that focus on determining the CLPs by observing the turning point at which the largest model impairment exists are misguided. The turning point actually represents the epoch when the model converges on defective data rather than indicating its CLPs. Given this understanding, to better track the CLPs, we should observe the effective gradient dynamics. However, the effective gradient is calculated based on the true gradient of the entire training data, which is practically inaccessible during the training process. Thus, our future work will explore the method for estimating the effective gradient and provide theoretical bounds for the approximation methods.

**Theoretical quantification of effective gradient deterioration under defective training.** Our analysis and the experiments shown in Figure 4 demonstrate that defective data can lead to a rapid deterioration of the effective gradient. However, we have yet to provide a theoretical framework to analyze such variation when model training with defective data. To rigorously quantify the permanent impairment caused by defective training, it is essential to account for the distribution shift and information missing introduced by defective data at the beginning of the training process, which will significantly drive the model optimization trajectory, leading to long-term consequences on model performance. In this paper, we still follow the current studies Achille et al. (2018); Yan et al. (2022) that primarily focus on the impact of blurred data which have shown the most pronounced effects, but fall short of providing comprehensive modeling of various forms of defective data. Addressing this gap requires developing a rigorous theoretical definition of the defective data. This is a challenging task that demands a deeper analysis of the interplay between data distribution shifts and gradient behavior during training. We plan to explore this direction in future research, building upon the effective gradient analysis foundations laid in our current exploration.

**Empowering attacks and defenses on DNNs with CLPs natures.** Since our exploration has theoretically demonstrated that training gains during CLPs can significantly impact overall model performance, it is worthwhile to extend this understanding to gain deeper insights and improve current attack and defense mechanisms in DNNs. However, existing explorations of CLPs-centered attacks and defenses Yan et al. (2023a;b) focus solely on the federated learning paradigm, proposing to allocate more attack/defense resources to the rounds during which the model is still in the CLPs. While these studies have provided valuable insights, a more in-depth analysis is needed to generalize this approach beyond federated learning and examine how CLPs dynamics can influence model vulnerabilities and robustness in broader training settings. Additionally, exploring the interactions between CLPs and different types of adversarial attacks, such as data poisoning, backdoor insertion, or model inversion attacks, could offer practical insights into improving model resilience. For instance, we could examine how poisoning attacks can be more effective during CLPs or how defenses could be optimized by identifying and mitigating vulnerabilities specific to these critical periods. Understanding these interactions will not only enhance model robustness but also lead to the development of more effective defense strategies tailored to CLPs-specific vulnerabilities. We aim to continue investigating these areas in future work.

## 6 Conclusion

In this study, we have critically revisited the CLPs and challenged existing FI-based explanations. We found the FI dynamics do not match the model memorization variation and CLPs do not center on the memorization phase as well. Hence, a new metric, the effective gradient, was proposed to explain the CLPs from the optimization perspective. Our study not only advances the exploration of CLPs but also highlights their significant implications for training robust neural networks.

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

# A MATHEMATICAL PROOFS

## A.1 GENERAL ASSUMPTIONS FOR SGD

Considering the SGD with the noisy gradient, in particular, we always hold the following assumptions (Nemirovski et al., 2009; Li & Orabona, 2019):

**A1**: The loss function is *L-Lipschitz*, i.e., $|\ell(\theta_1) - \ell(\theta_2)| \leq L||\theta_1 - \theta_2||, \forall \theta_1, \theta_2 \in \mathbb{R}^d$.

**A2**: The loss function is *M-smooth*, i.e., $\ell$ is differentiable and its gradient is *M-Lipschitz*: $||\nabla\ell(\theta_1) - \nabla\ell(\theta_2)|| \leq M||\theta_1 - \theta_2||, \forall \theta_1, \theta_2 \in \mathbb{R}^d$.

**A3**: The stochastic gradient $\nabla\ell(\theta; \xi)$ is the unbiased estimation of true gradient $\nabla\ell(\theta)$, i.e., $\mathbb{E}[\nabla\ell(\theta; \xi)] = \nabla\ell(\theta), \forall \theta \in \mathbb{R}^d$.

**A4**: The noise in the stochastic gradient is bounded by the noise variance $\sigma$, i.e., $E[||\nabla\ell(\theta; \xi) - \nabla\ell(\theta)||^2] \leq \sigma^2, \forall \theta \in \mathbb{R}^d$.

Note that the gradient is always calculated on the current model $\theta_t$ and data $\mathbf{x}$, so we simplify the heavy notation by omitting $\theta_t$ and $\mathbf{x}$ in $\nabla_{\theta_t}\ell(\mathbf{x}; \theta)$

## A.2 PROOF OF THEOREM 1

*Proof.* Since the loss function is M-smooth (Assumption A2), we have:

$$\ell(\theta_{t+1}) \leq \ell(\theta_t) + \nabla\ell(\theta_t)^\top(\theta_{t+1} - \theta_t) + \frac{M}{2}\|\theta_{t+1} - \theta_t\|^2 . \tag{9}$$

Given the SGD update rule: $\theta_{t+1} = \theta_t - \eta_t\nabla\ell(\theta_t; \xi_t)$, we substituting back it to Eqn. (9) and have

$$\ell(\theta_{t+1}) \leq \ell(\theta_t) - \eta_t\nabla\ell(\theta_t)^\top\nabla\ell(\theta_t; \xi_t) + \frac{M}{2}\eta_t^2\|\nabla\ell(\theta_t; \xi_t)\|^2 . \tag{10}$$

Given the assumption A3, we take the expectation over the randomness in noise $\xi_t$ and the conditioning on model parameters $\theta_t$ as follows

$$\mathbb{E}[\ell(\theta_{t+1})] \leq \mathbb{E}[\ell(\theta_t)] - \eta_t\mathbb{E}[\|\nabla\ell(\theta_t)\|^2] + \frac{M}{2}\eta_t^2\mathbb{E}[\|\nabla\ell(\theta_t; \xi_t)\|^2] . \tag{11}$$

From Assumption A4, we have:

$$\mathbb{E}[\|\nabla\ell(\theta_t; \xi_t)\|^2] = \mathbb{E}[\|\nabla\ell(\theta_t) + (\nabla\ell(\theta_t; \xi_t) - \nabla\ell(\theta_t))\|^2] \tag{12}$$
$$= \|\nabla\ell(\theta_t)\|^2 + \mathbb{E}[\|\nabla\ell(\theta_t; \xi_t) - \nabla\ell(\theta_t)\|^2]$$
$$\leq \|\nabla\ell(\theta_t)\|^2 + \sigma^2 .$$

So, we substitute Eqn. (12) back into the inequality Eqn. (11) to have

$$\mathbb{E}[\ell(\theta_{t+1})] \leq \mathbb{E}[\ell(\theta_t)] - \eta_t\mathbb{E}[\|\nabla\ell(\theta_t)\|^2] + \frac{M}{2}\eta_t^2\left(\|\nabla\ell(\theta_t)\|^2 + \sigma^2\right) . \tag{13}$$

Bring the terms involving $\mathbb{E}[\ell(\theta_{t+1})]$ to the left, we have

$$\eta_t\mathbb{E}[\|\nabla\ell(\theta_t)\|^2] \leq \mathbb{E}[\ell(\theta_t)] - \mathbb{E}[\ell(\theta_{t+1})] + \frac{M}{2}\eta_t^2\left(\|\nabla\ell(\theta_t)\|^2 + \sigma^2\right) . \tag{14}$$

Then, we sum both sides over $t = 1$ to $T$ as

$$\sum_{t=1}^{T}\eta_t\mathbb{E}[\|\nabla\ell(\theta_t)\|^2] \leq \mathbb{E}[\ell(\theta_1)] - \mathbb{E}[\ell(\theta_{T+1})] + \frac{M}{2}\sum_{t=1}^{T}\eta_t^2\left(\|\nabla\ell(\theta_t)\|^2 + \sigma^2\right) . \tag{15}$$

Since the loss function is bounded below (e.g., non-negative), we have:

$$\mathbb{E}[\ell(\theta_1)] - \mathbb{E}[\ell(\theta_{T+1})] \leq \mathbb{E}[\ell(\theta_1)] = L . \tag{16}$$

Then, we can simplify the inequality Eqn. (15) as

$$\sum_{t=1}^{T} \left( \eta_t - \frac{M}{2} \eta_t^2 \right) \mathbb{E}\left[ \|\nabla \ell(\theta_t)\|^2 \right] \leq L + \frac{M}{2} \sigma^2 \sum_{t=1}^{T} \eta_t^2 \ . \tag{17}$$

Since the learning rate $\eta_t < \frac{2}{M}$, we have

$$\eta_t - \frac{M}{2} \eta_t^2 \geq \frac{\eta_t}{2} \ . \tag{18}$$

Thus, for sufficiently large $t$:

$$\sum_{t=1}^{T} \frac{\eta_t}{2} \mathbb{E}\left[ \|\nabla \ell(\theta_t)\|^2 \right] \leq L + \frac{M}{2} \sigma^2 \sum_{t=1}^{T} \eta_t^2 \ . \tag{19}$$

Divide both sides by $\sum_{t=T_0}^{T} \frac{\eta_t}{2}$, we have:

$$\frac{\sum_{t=T_0}^{T} \eta_t \mathbb{E}\left[ \|\nabla \ell(\theta_t)\|^2 \right]}{\sum_{t=T_0}^{T} \eta_t} \leq \frac{2L}{\sum_{t=T_0}^{T} \eta_t} + M \sigma^2 \frac{\sum_{t=1}^{T} \eta_t^2}{\sum_{t=T_0}^{T} \eta_t} \ . \tag{20}$$

Taking the limit as $T \to \infty$, we have

$$\lim_{T \to \infty} \frac{\sum_{t=T_0}^{T} \eta_t \mathbb{E}\left[ \|\nabla \ell(\theta_t)\|^2 \right]}{\sum_{t=T_0}^{T} \eta_t} = 0 \tag{21}$$

Since the learning rates $\eta_t < \frac{2}{M}$ are positive, we have

$$\liminf_{t \to \infty} \mathbb{E}\left[ \|\nabla \ell(\theta_t)\|^2 \right] = 0 \ . \tag{22}$$

Finally, based on the definition of the trace of the Fisher Information Matrix: $\text{Tr}(I(\theta)) = \mathbb{E}\left[ \|\nabla \ell(\theta; \xi)\|^2 \right]$ and the unbiased estimation assumption A3, we have

$$\lim_{t \to \infty} \text{Tr}(I(\theta_t)) = 0 \ . \tag{23}$$

$\square$

### A.3 Proof of Lemma 1

*Proof.* We directly expand the left-hand side of the inequality, then we have

$$\begin{aligned}
\mathbb{E}[\|\nabla \ell(\theta; \xi)\|^2] &= \mathbb{E}[\|\nabla \ell(\theta; \xi) - \nabla \ell(\theta) + \nabla \ell(\theta)\|^2] \\
&= \mathbb{E}[\|\nabla \ell(\theta) + (\nabla \ell(\theta; \xi) - \nabla \ell(\theta))\|^2] \\
&= \mathbb{E}[\|\nabla \ell(\theta)\|^2] + 2\mathbb{E}[\langle \nabla \ell(\theta), \nabla \ell(\theta; \xi) - \nabla \ell(\theta) \rangle] \\
&\quad + \mathbb{E}[\|\nabla \ell(\theta; \xi) - \nabla \ell(\theta)\|^2] \ .
\end{aligned} \tag{24}$$

According to the unbiased estimation assumption **A3**, we have

$$\mathbb{E}[\nabla \ell(\theta; \xi) - \nabla \ell(\theta)] = \mathbb{E}[\nabla \ell(\theta; \xi)] - \mathbb{E}[\nabla \ell(\theta)] = 0 \ . \tag{25}$$

Thus, the second term in Eqn. (24) can be

$$2\mathbb{E}[\langle \nabla \ell(\theta), \nabla \ell(\theta; \xi) - \nabla \ell(\theta) \rangle] = 2\langle \nabla \ell(\theta), \mathbb{E}[\nabla \ell(\theta; \xi) - \nabla \ell(\theta)] \rangle = 0 \tag{26}$$

Besides, given the noise bound assumption **A4**, the third term in Eqn. (24) can be

$$\mathbb{E}[\|\nabla \ell(\theta; \xi) - \nabla \ell(\theta)\|^2 \leq \sigma^2 \tag{27}$$

As a result, Eqn. (24) can be

$$\begin{aligned}
\mathbb{E}[\|\nabla \ell(\theta; \xi)\|^2] &= \mathbb{E}[\|\nabla \ell(\theta)\|^2] + 2\mathbb{E}[\langle \nabla \ell(\theta), \nabla \ell(\theta; \xi) - \nabla \ell(\theta) \rangle] \\
&\quad + \mathbb{E}[\|\nabla \ell(\theta; \xi) - \nabla \ell(\theta)\|^2] \\
&\leq \mathbb{E}[\|\nabla \ell(\theta)\|^2] + \sigma^2
\end{aligned} \tag{28}$$

$\square$

## A.4 PROOF OF LEMMA 2

*Proof.* We first demonstrate that the model can converge given the learning rate. Since the stochastic gradients is the unbiased estimation, here, we omit the notation for the noise. Given **A1** and **A2**, we have the following equivalent expression based on Nesterov (2013),

$$\ell(\theta_2) - \ell(\theta_1) - \langle \nabla\ell(\theta_1), \theta_2 - \theta_1 \rangle \leq \frac{M}{2}||\theta_2 - \theta_1||^2 \ . \tag{29}$$

Considering $\theta_1 = \theta_i$ and $\theta_2 = \theta_{i+1}$, we have

$$\ell(\theta_{i+1}) - \ell(\theta_i) - \langle \nabla\ell(\theta_i), \theta_{i+1} - \theta_i \rangle \leq \frac{M}{2}||\theta_{i+1} - \theta_i||^2 \ . \tag{30}$$

We reorganize the above inequality and consider the Eqn. (4),

$$
\begin{aligned}
\ell(\theta_{i+1}) \leq &\ell(\theta_i) + \langle \nabla\ell(\theta_i), \theta_{i+1} - \theta_i \rangle + \frac{M}{2}||\theta_{i+1} - \theta_i||^2 \\
\leq &\ell(\theta_i) - \eta\langle \nabla\ell(\theta_i), \nabla\ell(\theta_i) \rangle + \frac{M}{2}\eta^2||\nabla\ell(\theta_i)||^2 \\
\leq &\ell(\theta_i) - \eta||\nabla\ell(\theta_i)||^2 + \frac{M}{2}\eta^2||\nabla\ell(\theta_i)||^2 \\
\leq &\ell(\theta_i) - \eta(1 - \frac{M\eta}{2})||\nabla\ell(\theta_i)||^2
\end{aligned}
$$

Since $\eta \leq \frac{2}{M}$, $\eta(1 - \frac{M\eta}{2}) \geq 0$. Therefore, we have

$$\ell(\theta_{i+1}) \leq \ell(\theta_i) \ . \tag{31}$$

It indicates that if learning rate $\eta \leq \frac{2}{M}$, the loss function can converge, resulting in the fact that the real gradient should converge to 0, that is

$$\mathbb{E}[||\nabla\ell(\theta)||^2] \to 0. \tag{32}$$

Then, we bring it to Lemma 1, we have

$$\mathbb{E}[||\nabla\ell(\theta; \xi)||^2] \leq \mathbb{E}[||\nabla\ell(\theta)||^2] + \sigma^2 \to \sigma^2. \tag{33}$$

$\square$

## A.5 PROOF OF THEOREM 2

*Proof.* Based on the definition of effective gradient, its norm can be expressed as

$$
\begin{aligned}
||\hat{\ell}(\theta; \xi)||^2 = &\left|\left| Proj_{\ell(\theta)}\ell(\theta; \xi) \right|\right|^2 \\
= &\left|\left| \frac{\nabla\ell(\theta; \xi) \cdot \nabla\ell(\theta)}{||\nabla\ell(\theta)||^2} \right|\right|^2 ||\nabla\ell(\theta)||^2
\end{aligned}
$$

Take the expectation on both sides of the formula, we have

$$\mathbb{E}[||\hat{\ell}(\theta; \xi)||^2] = \mathbb{E}\left[ \left|\left| \frac{\nabla\ell(\theta; \xi) \cdot \nabla\ell(\theta)}{||\nabla\ell(\theta)||^2} \right|\right|^2 ||\nabla\ell(\theta)||^2 \right]$$

Based to the unbiased estimation assumption **A3**, we have

$$\mathbb{E}[\nabla\ell(\theta; \xi)] = \mathbb{E}[\nabla\ell(\theta)] = \nabla\ell(\theta) \ . \tag{34}$$

Thus, we have

$$\mathbb{E}\left[ \left|\left| \frac{\nabla\ell(\theta; \xi) \cdot \nabla\ell(\theta)}{||\nabla\ell(\theta)||^2} \right|\right|^2 \right] = 1$$

Then, given Lemma 2, the expected norm of effective gradient can be

$$\mathbb{E}[||\hat{\ell}(\theta; \xi)||^2] = \mathbb{E}[||\nabla\ell(\theta)||^2] \to 0$$

$\square$

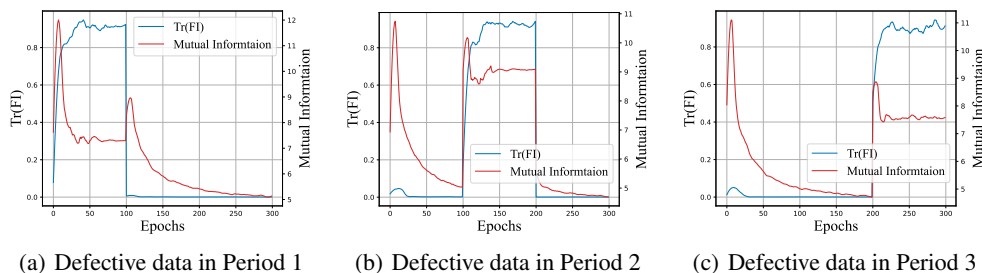

(a) Defective data in Period 1     (b) Defective data in Period 2     (c) Defective data in Period 3

Figure 5: The mutual Information and the FI variations on VGG-16.

# B  EXPERIMENT SUPPLEMENTS

## B.1  EXPERIMENT SETTINGS

In all the experiments, we employ the standard model structure and implementation for VGG-16 (Simonyan & Zisserman, 2014), ResNet-18 (He et al., 2016), and DeiT (Touvron et al., 2021). Unless otherwise specified, the optimizer is set to SGD with a fixed learning rate. Besides, We employ standard data augmentation with random horizontal flipping and Random affine to mitigate overfitting. To create the defective blurred data, we use the GaussianBlur function with a strong parameter of kernel size=7 and sigma=3. For VGG-16 and ResNet-18, learning rate is $0.01$, weight decay is $1e^{-6}$, the batch size is $256$. For DeiT, the learning rate is $0.05$, weight decay is $1e^{-6}$, the batch size is $256$. Besides, the channel size is 3, the patch size is 4, the embed size is 512, the number of heads is 8, the number of heads is 4, and the hidden size is 512. The teacher model is trained by VGG-16. Our experiments are conducted on a workstation equipped with NVIDIA RTX 4090 GPU with 24GB of VRAM.

## B.2  METRICS CALCULATION

### B.2.1  APPROXIMATION OF THE TRACE OF FISHER INFORMATION MATRIX

The trace of the Fisher Information Matrix (*FIM*) is calculated based on its definition as

$$Tr(F(\theta)) = \mathbb{E}_{x \sim \hat{Q}(x)} \left[ \mathbb{E}_{y \sim p_\theta(y|x)} \left[ ||\nabla_\theta \log p_\theta(y|x)||^2 \right] \right] , \tag{35}$$

where $x$ is sampled based on the Monte-Carlo methods in the training dataset $X$ and $y$ is the corresponding label.

### B.2.2  APPROXIMATION OF THE MUTUAL INFORMATION

To calculate the mutual information between training data and the model output representation, we employ the binning methods following these steps:

First, for each class, we discretize continuous model outputs by binning the logits for each class into equal intervals between -1 and 1, making it easier to calculate joint and marginal probabilities. For example, according to 30 equal intervals, we have the bins $\{-1, -0.933, \cdots, 0.933, 1\}$. Then the logits $t = [-0.9, -0.5, 0.1, 0.4, 0.8]$ can be transferred to $[1, 7, 16, 21, 27]$. Second, we create a contingency table that counts the occurrences of each combination of training data and model output bins to calculate their joint distribution $P(x, t)$. Then, we compute the marginal distributions $P(x)$ and $P(t)$ by summing the counts over the rows and columns of the contingency table, respectively. Third, we use these joint and marginal distributions to calculate mutual information based on the definition:

$$I(X; T) = D_{KL}[p(x, t) \| p(x)p(t)] .$$

Here, the number of intervals is defined as the hyperparameter, which will influence the approximation performance. We follow Shwartz-Ziv & Tishby (2017) to set it as 30.

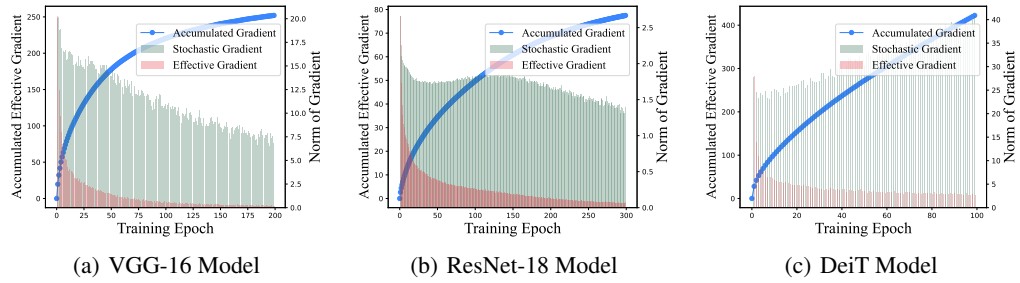

|  |  |  |
|---|---|---|
| (a) VGG-16 Model | (b) ResNet-18 Model | (c) DeiT Model |

Figure 6: The variation of the norm of effective gradient, stochastic gradient, and the accumulated effective gradient with normal training of SGD with momentum=0.9.

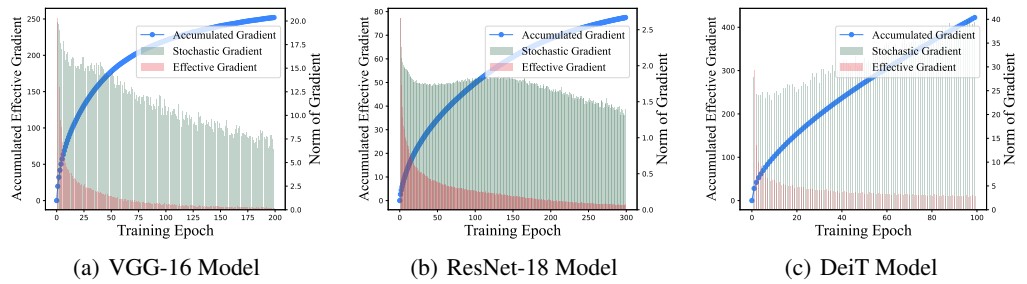

|  |  |  |
|---|---|---|
| (a) VGG-16 Model | (b) ResNet-18 Model | (c) DeiT Model |

Figure 7: The variation of the norm of effective gradient, stochastic gradient, and the accumulated effective gradient with normal training of SGD with Nesterov.

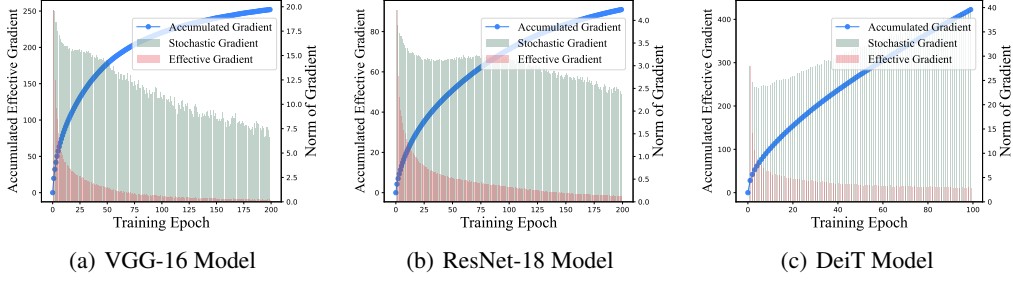

|  |  |  |
|---|---|---|
| (a) VGG-16 Model | (b) ResNet-18 Model | (c) DeiT Model |

Figure 8: The variation of the norm of effective gradient, stochastic gradient, and the accumulated effective gradient with normal training of Adam.

## B.3   Additional Plots

This section presents the plots for additional experiments.

First, we show the additional results referred to in Section 3. Figure 1 exhibits the variations in mutual information and FI on VGG-16. These figures display the same phenomenon observed with ResNet-18. Additionally, we present the results referred to in Section 4. Here, we train models with the optimizers: SGD with momentum, SGD with Nesterov, and Adam. Figures 6, 7, and 8 show the variations in the norms of the effective gradient, the stochastic gradient, and the accumulated effective gradient, respectively. The effective gradients exhibit a decreasing trend in all scenarios.

