# OpenReview forum: "Revisiting Critical Learning Periods in Deep Neural Networks"
_ICLR.cc/2025/Conference — ICLR 2025 Conference Withdrawn Submission_

### Official Review · Reviewer_4JsX · 2024-10-31

**Soundness:** 3
**Presentation:** 3
**Contribution:** 3
**Rating:** 6
**Confidence:** 3

**Summary:**

The submission investigates critical learning periods (CLP), in which models can be permanently impaired by defective data. The authors refute the previous explainer that relied on FI and related the impairment to memorization. Instead, they develop 'effective gradients' as a metric to identify updates in which the model learns meaningful information. They demonstrate that effective gradients are well-suited to explain CLPs, while FI cannot.

**Strengths:**

- The paper considers an important problem of understanding learning dynamics and critical phases of neural networks.
- The manuscript is well-written and largely builds arguments nicely.
- The two main arguments that FI is a symptom of a learning phase not the cause as well as effective gradients as better CLP explainer are well supported, both by theory and empirical evidence.
- The findings of the submissions shed light on learning phases. The may help train models more efficiently, e.g. by understanding at what training phase data has to be cleaner, which has implications for data curation but also augmentation.

**Weaknesses:**

- While I appreciate the notion of the effective gradient, the derivation of it takes up a lot of space in the paper. Effectively, this could have been a simple statement of 'alignment between current gradient and $(w_final - w_i)$. As far as I can tell, the takeaway from the theory is that the effective gradient converges 0 while the actual gradient doesn't, hence while the effective gradient is large the model can be tripped up by bad data? In that case, it might help to start with a motivation / summary of the argument rather than building the argument up from the parts.
- The effective gradient provides good grounding for the experiments in this paper, but it's only available for post-hoc analysis. It seems to me that it might be correlated to per-step loss reduction, albeit not perfectly. Metrics like these might be measurable while training and provide actionable insights at what training phase to be extra careful about data quality.
- The experimental evaluation clearly supports the main argument of the paper. However, some details are unclear. For instance, the 'optimum' used to compute the effective gradient, is it the same between the different models, or do they each get their own, individual optima? Also, the claim that models with defective training reach a different basin would benefit from more support, e.g. by computing the mode connectivity between the clean/defective solution.
- The paper is generally well-written and carries a clear message. However, there seems to be some unnecessary math like on page 2 the KL. In the interest of clarity, I would recommend dropping it unless it helps make the argument.

**Questions:**

See weaknesses.

---

### Official Review · Reviewer_uwEW · 2024-11-01

**Soundness:** 2
**Presentation:** 3
**Contribution:** 3
**Rating:** 5
**Confidence:** 3

**Summary:**

This paper revisits the critical learning periods (CLPs) in training deep networks and challenges the existing, prevalent Fisher Information (FI)-based explanations. The authors first highlight the misalignment between CLPs and FI dynamics due to defective data in the early training stages. To enhance the understanding of CLPs, the authors propose a novel metric called effective gradients and investigate its relationship with CLPs. Experiments on various popular deep learning models support the proposed theories. Several key findings are summarized based on the experimental results, and providing readers with novel insights into CLPs, along with other detailed discussion and analysis.

**Strengths:**

(1) This paper develops solid theories and methods, supported by experiments, to demonstrate that the CLPs period may not necessarily align with, nor be explained by, certain patterns in FI dynamics.

(2) The proposed metric of effective gradients offers meaningful insights into the training process of deep learning models optimized by stochastic gradient descent, making it a great fit for investigating and explaining CLPs. Key findings derived from effective gradients provide valuable and novel perspectives on understanding CLPs, and it may effectively inspire future research that aligns with the interests of the ICLR community.

**Weaknesses:**

(1) The experiments lack statistical analysis of the variance in metrics and model performance during CLPs, especially given that it is the "stochastic" gradient descent that behinds the research problem and motivates the proposed novel metrics. The authors are encouraged to add variance plots to the figures and conduct statistical tests to help readers better evaluate the consistency of the key findings.

(2) The explanation of experiment results regarding the relationship between FI dynamics and the memory phase seems questionable. In both Fig. 1 (b)(c), there is a slight initial increase followed by a decrease in FI at the beginning of training, although the change is not as large as in the defective training periods. Do these small changes of FI, at the beginning of the training, align well with the existing explanations of memorization and forgetting phases? Can one then simply conclude that "the surge in FI is a consequence of defective training"? The authors are encouraged to provide more explanations and discussion on this.

(3) Given the challenge of precisely defining CLPs in the training of deep learning models, the authors are encouraged to provide a clear problem setting for the CLPs that this paper aims to investigate and address, ideally in the first paragraph of the introduction. Personally, I found myself a bit lost in Sections 2.2 and 2.3, trying to understand what CLPs should be without considering FI dynamics, and then gained a much clearer understanding when I reached the discussion section, particularly the paragraph, "Critical learning periods are not time windows with clear boundaries." Additionally, the authors mentioned "The CLPs refer to the phenomenon where ..." in Section 4.2 but did not provide references there. In summary, previewing some key findings (like potentially a new "definition" and "understanding" of CLPs) in the early part of the introduction would help strengthen the writing and clarify the definition of CLPs in this paper.

**Questions:**

(1) Since defective data is also "new" data, how can one distinguish between the different impacts introduced by "new" data versus "defective" data? Did the authors conduct any experiments or have any insight on what the effective gradients would look like when training on new but high-quality data (where there is a domain shift but no concerns about data quality)?

---

> ### Comment · Reviewer_uwEW · 2024-11-26
>
> It seems the authors did not provide any response to the reviewers. I will maintain my score for now.

---

### Official Review · Reviewer_qETJ · 2024-11-02

**Soundness:** 2
**Presentation:** 2
**Contribution:** 1
**Rating:** 3
**Confidence:** 2

**Summary:**

The paper studies the validity of existing explanation via Fisher Information (FI) metric to quantify critical learning periods (CLPs) phenomenon. The paper uses mutual information (MI) between the input and neural representation as an alternate metric for memorization and studies the dynamics of MI and FI. Empirically, the paper shows that CLPs can extend beyond memorization phase as quantified via MI. The paper introduces a new metric called Effective Gradient that is optimization-based that shows better sensitivity to training under noisy (or defective) inputs.

**Strengths:**

- The paper studies CLPs which is an phenomenon that is of interest to the research community (as evidenced by citations to Achille et al.'s work)
- Training dynamics as quantified by mutual information (MI) and Fisher Information (FI) is interesting in the setting covered in the paper
- The paper proposes effective gradient, a new metric to explain CLPs. A new metric is commendable especially given its convergence behavior (Theorem 2) but please see weakness for concerns

**Weaknesses:**

- The paper does not define memorization clearly but starts using this term. This vague use can be very confusing to the reader. A suggested place to consider to define memorization is at the start of Section 2.

- The paper uses mutual information (MI) to measure memorization in Section 3.2. It would be beneficial to define this even before the mathematical preliminaries in prior subsections (also see first point). More importantly, there is no justification on why MI is a good definition of memorization in neural networks. A quick read of one of the references (Kleinman et al.)[https://arxiv.org/abs/2010.02459] appears to suggest that other/better metrics are preferred over MI to quantify memorization. The paper could benefit from a discussion of why MI is preferred in the current work over other metrics.

- (continuing with memorization metrics) There is vast literature that appears to define and use memorization measures some of which relates to noisy labels while others may related to noisy inputs. The paper would benefit if the paper includes a proper discussion of prior work to help both new and experienced readers to understand the current work's viewpoint

- There are other references that talk about memorization and Fisher Information (FI) that needs to be discussed in the paper. See (Jastrz ̨ebski et al.)[https://arxiv.org/abs/2012.14193] for instance. These prior works suggest that trace of FIM is good enough memorization measure. A discussion on preferring MI over other proposed measures could be useful.

- Certain sections on the paper are hard to follow. For instance it is hard for me to pick up the explanation in Sections 2.2 and 2.3 on FI and memorization.

- The paper proposes effective gradient. However, this metric is non-measurable so an approximation is used. However, the paper uses an approximation that relies on knowing the final solution (weights) which severely reduces the usefulness of this quantity both in concept as well as practice. The paper notes that the effective gradient is ``practically inaccessible''. However, the lack of access to effective gradient is not just in practice but also in theory in the case of online learning-like settings. A clarification on what settings effective gradient applies to would help clarify claims made in the paper.

**Questions:**

- Why is mutual information (MI) a good metric to characterize memorization? Especially in light of prior works that shows other metrics maybe more useful. An intuitive notion of memorization is to perfectly reconstruct input from embeddings. However, its not clear if the representations considered in the study can perfectly construct the input.
- Please consider discussing more works on memorization in the paper as noted in Weaknesses (No need for a  full list as the literature appears to be vast but important works such on noisy label memorization and/or noisy input memorization would be useful to general readers)
- The arguments in Section 2.2 and 2.3 may need to be simplified or expanded so that a general audience may appreciate the points raised by the authors especially in light of prior work (Jastrz ̨ebski et al.)[https://arxiv.org/abs/2012.14193] that appears to suggest trace of FI is a good metric for memorization
Overall, my decision to not support acceptance at this point is due to the concerns noted above and in the Weakness section. I look forward to a discussions with the authors to clear up any misunderstandings on my part.

---

### Official Review · Reviewer_tidT · 2024-11-04

**Soundness:** 3
**Presentation:** 2
**Contribution:** 2
**Rating:** 5
**Confidence:** 3

**Summary:**

This paper revisits and challenges the existing understanding of Critical Learning Periods (CLPs) in Deep Neural Networks. The authors demonstrate that the prevalent explanation of CLPs based on Fisher Information (FI) dynamics and model memorization is inaccurate. Traditionally, FI has been used to explain CLPs by correlating high FI values with increased model memorization during early training stages. However, the authors argue that such interpretations are flawed, as FI merely indicates sensitivity to training data and is influenced by noise rather than accurately representing memorization or CLPs. Through theoretical analysis and empirical results, the authors introduce a new metric - the effective gradient, to demonstrate that early training epochs disproportionately impact final performance due to their effect on model optimization dynamics.

**Strengths:**

- The authors present a well-supported critique of FI-based explanations, followed by a thorough theoretical framework for understanding CLPs from an optimization standpoint.
- The paper is generally clear, with a logical flow that makes the theoretical ideas accessible.
- Provides new theoretical framework for understanding CLPs

**Weaknesses:**

1. All experiments were conducted only on CIFAR-10 dataset, without validation on larger-scale datasets like ImageNet. Also, there is no exploration of domains beyond image classification. Testing on more complex datasets and more types of defective data could further validate the generality of the effective gradient metric. Do the authors expect their findings to generalize to larger datasets or other domains?

2. The calculation of effective gradients, as discussed, relies on access to the full gradient, which is challenging to obtain in real-time during training. Although the paper acknowledges this limitation, it lacks concrete solutions or approximations that could make the effective gradient metric feasible in practical scenarios. How would the authors propose to calculate effective gradients during actual training, rather than post-hoc? Or is it true that effective gradient can only be used as a post-training analysis tool and not for actual guidance in training? Could the authors discuss any existing methods for approximating full gradients that might be applicable, or outline potential research directions for developing such approximations?

3. The paper claims to "shed light on enhancing model performance and robustness through such periods" in the abstract. Could the authors provide examples or guidelines on how this understanding of CLPs could be used to improve training strategies?

4. What are the computational requirements for calculating effective gradients in large-scale training scenarios? Could you provide a complexity analysis? Or could you provide an estimate of the additional computational overhead (e.g., in terms of time or memory) required to calculate effective gradients for a typical training run on your experimental setup?

I'd be happy to raise the score if the author could address my concerns.

**Questions:**

See the weaknesses part.

---

### Note · Authors · 2025-04-23

I have read and agree with the venue's withdrawal policy on behalf of myself and my co-authors.

---

### Meta-Review · Area_Chair_7Vi3 · 2024-12-20

**Metareview:**

The paper contributes to the understanding of the critical learning periods in deep neural networks. Reviewers had certain significant comments (e.g. regarding the connection between FI dynamic and memorization) that were not addressed during the rebuttal phase. As such, I must recommend rejection at this stage. Thank you for submitting to ICLR and I hope comments will help you improve the work.

**Additional Comments On Reviewer Discussion:**

No further comments. Authors have not responded to comments and there was no discussion.

---

### Decision · Program_Chairs · 2025-01-22

Reject